# How Do Disaster Relief Nurses in Japan Perceive and Respond to Risks? A Cross-Sectional Study

Aki Nishikawa [1], Takumi Yamaguchi [2,3,4,*], Yumiko Yamada [5], Hideko Urata [6], Tetsuko Shinkawa [6]
and Yuko Matsunari [4]

[1] Nagasaki Rosai Hospital, Nagasaki 857-0134, Japan; akinsaki@nagasakih.johas.go.jp
[2] Research Administration Center, Saitama Medical University, Saitama 350-0495, Japan
[3] Nuclear Safety Research Association, Tokyo 105-0004, Japan
[4] School of Health Sciences, Kagoshima University, Kagoshima 890-8544, Japan;
matsuy@health.nop.kagoshima-u.ac.jp
[5] Department of Nursing, Kwassui Women's University, Nagasaki 856-0835, Japan; y.yamada@kwassui.ac.jp
[6] Atomic Bomb Disease Institute, Nagasaki University, Nagasaki 852-8523, Japan;
hideko@nagasaki-u.ac.jp (H.U.); tetuko@nagasaki-u.ac.jp (T.S.)
* Correspondence: takumi@saitama-med.ac.jp

**Abstract:** This study assessed the risk perceptions among disaster relief nurses (DRNs) in Japan by focusing on 15 risk factors associated with frequent natural disasters and the ongoing coronavirus disease 2019 (COVID-19) pandemic. We conducted a cross-sectional study that targeted DRNs across six prefectures in Japan and explored nurses' perceptions of risks including radiation exposure, volcanic eruptions, and mass infections. The findings indicated a heightened perception of radiation and nuclear-related risks. In the context of the COVID-19 pandemic, DRNs perceived "mass infection" as a significant risk. An age-based analysis revealed that younger nurses had more dread about "mass infection" and had heightened uncertainty about the "X-ray test" compared with their older peers. Understanding DRNs' risk perceptions is crucial for effective disaster response preparedness and training. The study highlights the need to address these perceptions to ensure that DRNs are well prepared and supported in their roles. This study was not pre-registered on a publicly accessible registry.

**Keywords:** disaster relief nursing; risk perception; disaster nursing

## 1. Introduction

Disaster relief nursing, a specialized field within healthcare, plays a pivotal role in responding to the immediate and long-term health needs of communities that are affected by disasters [1]. In Japan, the initiative to foster disaster relief nurses (DRNs) was launched following the Great Hanshin Earthquake in 1995 [2]. This initiative is particularly crucial in Japan, a country located in the "Ring of Fire"—an area known for its high tectonic activity and susceptibility to frequent and severe natural disasters including earthquakes, tsunamis, typhoons, and volcanic eruptions [3].

Risk perception, defined as an individual's subjective judgment about the likelihood of negative occurrences, is a critical factor in healthcare decision making and behavior [4]. In addition, risk perception is determined by two factors: the dread risk factor and the unknown risk factor [5]. For healthcare professionals, especially those who work in high-risk fields such as disaster relief, an accurate perception of risk can considerably impact the effectiveness of their response and the outcomes for those to whom they provide care [4].

The importance of understanding risk perception among healthcare workers is not limited to Japan and has been studied in various international contexts. For instance, a study that aimed to enhance vaccination rates in Bangladesh focused on seasonal influenza vaccine uptake among healthcare workers [6]. In Ethiopia, a cross-sectional survey that

assessed healthcare workers' knowledge and risk perception of coronavirus disease 2019 (COVID-19) found that approximately 58% perceived COVID-19 as a high-risk disease [7]. The UK-REACH (The United Kingdom Research study into Ethnicity And COVID-19 outcomes in Healthcare workers) project in the United Kingdom investigates the impact of ethnicity on COVID-19 diagnosis and clinical outcomes among healthcare workers and includes the risk perceptions of participants [8]. Studies from Australia and Brazil have also explored the drivers of COVID-19 vaccination and the mental health of healthcare workers, respectively [9]. These international studies underscore the universal relevance of healthcare workers' risk perception and its impact on healthcare outcomes. However, studies specifically focused on disaster-related risk perceptions among healthcare workers are lacking. Understanding these perceptions could facilitate more effective disaster relief efforts.

The aftermath of the 2011 Great East Japan Earthquake and the subsequent tsunami, one of the most devastating disasters in recent history, underscored the importance of an effective disaster response, which includes skilled disaster relief nursing [1]. Studies conducted in the aftermath of this disaster have highlighted the psychological impact on disaster relief workers and the notable prevalence of post-traumatic stress disorder and depression [1]. Furthermore, the Fukushima nuclear disaster that followed the earthquake and the disaster's significant effects on the mental health of the affected population have brought radiation risk into sharp focus [4].

However, despite these findings, a gap exists in our understanding of how DRNs, who are often at the forefront of such crises, perceive the risks associated with their work. This gap is particularly pronounced in the context of Japan because of the country's unique disaster profile. Understanding nurses' risk perception is crucial because it can influence their decision making, their mental health, and ultimately, their ability to provide effective care during disasters [4].

This study's purpose was to explore and understand risk perception among DRNs in Japan. By gaining insight into how these nurses perceive and respond to the risks associated with their work, we identified areas for improvement in training and support to ultimately enhance the effectiveness of disaster responses in Japan [10]. This study is unique in its focus on DRNs in Japan, a context that has not been extensively investigated in risk perception studies. The findings of this study can significantly contribute to our understanding of risk perception in disaster relief nursing and inform strategies to improve disaster preparedness and response in Japan and beyond.

## 2. Materials and Methods

### 2.1. Study Design and Setting

This was a cross-sectional study. We targeted DRNs in Japan who were registered in Fukushima, Saga, Nagasaki, Kumamoto, Oita, and Miyazaki prefectures. Other than Fukushima, these prefectures are located in the Kyushu region, which has experienced over 70 disasters in the past 30 years, making it one of the most disaster-prone regions in Japan. This region was selected because an assessment of the risk perception among DRNs who work in such disaster-prone areas can contribute significantly to more effective disaster responses in the future.

This study was conducted from July to September 2020.

### 2.2. Participants

A survey request document and a consent form were mailed to the president of each prefectural nursing association. The number of DRNs who agreed to participate in the survey in each prefecture was noted. Subsequently, a document requesting survey cooperation and a letter explaining the URL and QR code of the survey form were mailed to the nursing association in each prefecture. Next, the designated individual from each prefectural nursing association dispatched a written survey request to the DRNs accompanied by an explanatory document that contained a URL and QR code. The dispatch was either direct

or through the facilities in which the nurses were based, provided that the facilities had agreed to participate in the survey.

The online survey was created using Google Forms. After the participants responded to the web-based survey, the data were collected by the researcher. The completed web-based surveys were then sent to the researcher for data collection.

In total, 466 of the 986 nurses (47.2%) answered the questionnaire. After excluding 65 nurses who failed to respond to the risk perception question items, 401 respondents were included in the analysis (Figure 1).

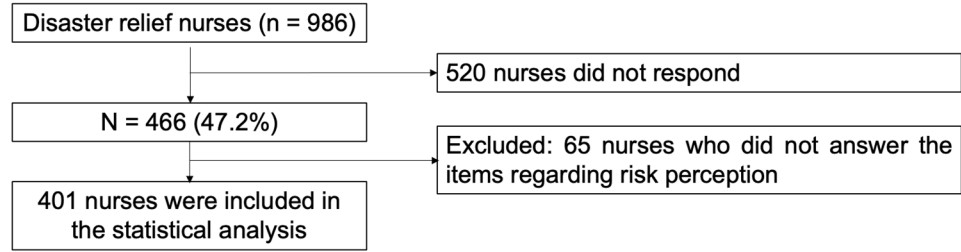

**Figure 1.** Enrollment of study participants.

### 2.3. Variables

The survey items were formulated based on previous studies [5,11–13]. Demographic factors such as gender, age, and job title were considered. Additionally, participants' perceptions of 15 risks were surveyed. The risks were (1) radiation exposure, (2) volcanic eruption, (3) floods, (4) norovirus, (5) radioactive contamination, (6) nuclear weapons (nuclear war), (7) mass infection, (8) Ebola, (9) car accident, (10) X-ray test, (11) dengue fever, (12) cervical cancer vaccine, (13) nuclear reactor accident, (14) tobacco, and (15) genetic engineering.

The participants rated their perception of each risk on a 7-point scale, with options ranging from "not at all" to "very strongly". The items were categorized as "dread risk" or "unknown risk".

### 2.4. Statistical Methods

Initially, the scores for each risk factor were based on "dread risk" and "unknown risk", and their means were calculated. The coordinates of each factor were then plotted on a graph, with "dread risk" on the horizontal axis and "unknown risk" on the vertical axis.

To evaluate the distribution of the scores, the Shapiro–Wilk test was employed to assess parametricity [14]. This non-parametric test is widely used for assessing non-parametric data. Given that the distribution was non-parametric, a Jonckheere–Terpstra test was conducted to compare the differences in "dread risk" and "unknown risk" scores across age groups [15,16]. This test is a non-parametric method for ordered alternatives and is suitable for analyzing ordinal data.

To further validate the robustness of our findings, a post hoc sample size calculation was performed using a bootstrap-based Jonckheere–Terpstra test. Specifically, the test was applied to 1000 bootstrap samples for each risk factor. Approximately 96.5% of the bootstrap samples resulted in a $p$-value less than or equal to 0.05. This high proportion indicated that the sample size provided sufficient statistical power to detect significant differences across age groups at the 0.05 significance level.

All statistical analyses were performed using R version 4.3.3, and the $p$-value was set at 0.05.

### 3. Results

#### 3.1. Descriptive Statistics

The descriptive statistics are presented in Table 1. Women represented 82.5% of the respondents, and the largest age group was those in their 40s (44.6%). The most common job title was "staff nurse", which accounted for 61.6% of the respondents.

**Table 1.** Descriptive statistics of study participants.

| Variables | | n | % |
|---|---|---|---|
| Gender | | | |
| | Male | 70 | 17.5 |
| | Female | 331 | 82.5 |
| Age (years) | | | |
| | 20s | 9 | 2.2 |
| | 30s | 105 | 26.2 |
| | 40s | 179 | 44.6 |
| | 50s | 91 | 22.7 |
| | ≥60s | 17 | 4.2 |
| Job Title | | | |
| | Staff | 247 | 61.6 |
| | Deputy Chief | 80 | 20.0 |
| | Chief | 46 | 11.5 |
| | Director | 15 | 3.7 |
| | Certified Nurse | 3 | 0.7 |
| | Other | 10 | 2.5 |

*3.2. Risk Perception Mapping*

Scores for each risk factor were based on "dread risk" and "unknown risk", and their means were calculated. The coordinates of each factor were plotted on a graph, with "dread risk" on the horizontal axis and "unknown risk" on the vertical axis. The coordinates of each risk factor were as follows: (1) radiation exposure (x, y = 0.110, 0.434), (2) volcanic eruption (−0.100, 0.364), (3) floods (0.075, 0.833), (4) norovirus (−0.840, −0.209), (5) radioactive contamination (0.147, 0.524), (6) nuclear weapons (nuclear war; 0.332, 0.681), (7) mass infection (0.107, 0.883), (8) Ebola (0.015, 0.299), (9) car accident (−0.673, 0.027), (10) X-ray test (−1.377, −1.579), (11) dengue fever (−0.374, −0.377), (12) cervical cancer vaccine (−0.945, −1.222), (13) nuclear reactor accident (0.319, 0.586), (14) tobacco (−0.900, −0.364), and (15) genetic engineering (−0.229, −0.708). The results of the risk mapping are shown in Figure 2.

*3.3. Differences in Risk Scores by Age Group*

The risk perceptions for "dread risk" across the age groups were analyzed for the 15 risk factors. The perception of "mass infection" was statistically different between the age groups ($p$ = 0.037; Table 2).

**Table 2.** Differences in "dread risk" scores by age group (mean ± standard deviation).

| Age Group | 20s (n = 9) | 30s (n = 105) | 40s (n = 179) | 50s (n = 91) | ≥60s (n = 17) | *p*-Value |
|---|---|---|---|---|---|---|
| Radiation Exposure | 4.11 ± 1.91 | 4.21 ± 1.65 | 4.50 ± 1.49 | 4.52 ± 1.55 | 4.88 ± 1.28 | 0.08 |
| Volcanic Eruption | 4.44 ± 1.17 | 4.47 ± 1.53 | 4.28 ± 1.53 | 4.30 ± 1.42 | 4.88 ± 1.02 | 0.569 |
| Flood | 4.56 ± 0.83 | 4.98 ± 1.06 | 4.80 ± 1.07 | 4.73 ± 1.20 | 4.94 ± 1.06 | 0.301 |
| Norovirus | 3.33 ± 0.83 | 3.76 ± 1.44 | 3.83 ± 1.42 | 3.89 ± 1.36 | 3.24 ± 1.63 | 0.685 |
| Radioactive Contamination | 4.56 ± 1.17 | 4.27 ± 1.57 | 4.59 ± 1.46 | 4.67 ± 1.28 | 4.65 ± 1.08 | 0.136 |
| Nuclear Weapons (Nuclear War) | 5.00 ± 0.82 | 4.46 ± 1.87 | 4.69 ± 1.75 | 4.84 ± 1.61 | 4.94 ± 1.30 | 0.148 |
| Mass Infection | 4.67 ± 0.67 | 5.07 ± 1.12 | 4.85 ± 1.04 | 4.82 ± 1.12 | 4.53 ± 0.92 | 0.037 |
| Ebola | 4.78 ± 0.92 | 4.16 ± 1.93 | 4.33 ± 1.88 | 4.31 ± 1.68 | 4.53 ± 1.33 | >0.999 |
| Car Accident | 4.33 ± 1.05 | 4.15 ± 1.41 | 3.99 ± 1.39 | 3.89 ± 1.45 | 4.18 ± 0.98 | 0.184 |
| X-ray Test | 2.89 ± 1.37 | 2.72 ± 1.66 | 2.26 ± 1.57 | 2.40 ± 1.43 | 2.12 ± 1.18 | 0.068 |
| Dengue Fever | 4.22 ± 0.79 | 3.50 ± 1.91 | 3.66 ± 1.80 | 3.62 ± 1.55 | 3.65 ± 1.71 | 0.914 |
| Cervical Cancer Vaccine | 3.44 ± 1.34 | 2.92 ± 1.49 | 2.55 ± 1.58 | 3.01 ± 1.37 | 2.71 ± 1.49 | 0.782 |
| Nuclear Reactor Accident | 4.67 ± 0.94 | 4.31 ± 1.80 | 4.60 ± 1.83 | 4.80 ± 1.48 | 4.94 ± 1.30 | 0.056 |
| Tobacco | 3.89 ± 1.37 | 3.56 ± 1.49 | 3.52 ± 1.56 | 3.88 ± 1.24 | 3.88 ± 1.08 | 0.304 |
| Genetic Engineering | 4.00 ± 0.67 | 3.17 ± 1.75 | 3.17 ± 1.62 | 3.51 ± 1.36 | 3.82 ± 0.98 | 0.269 |

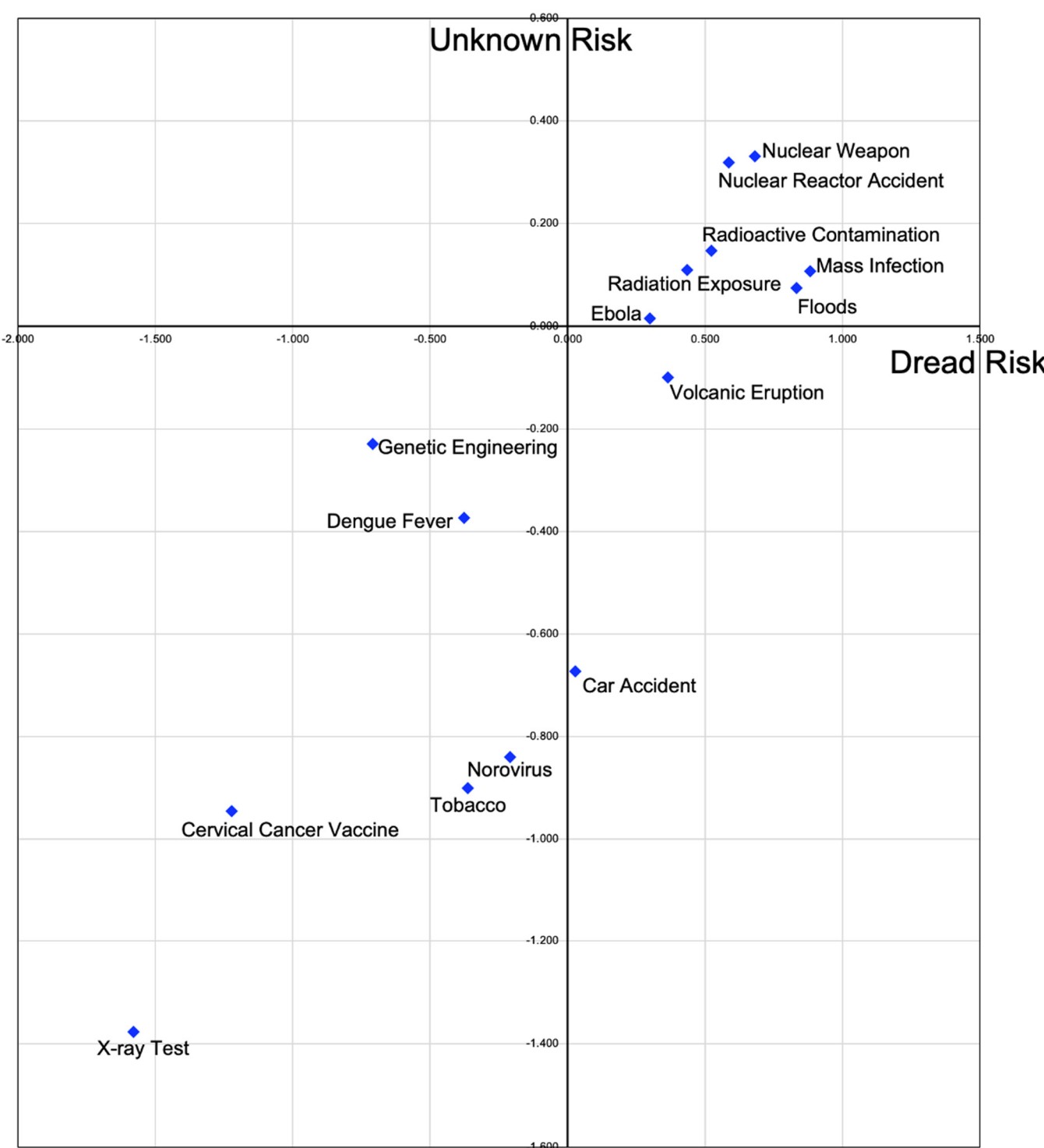

**Figure 2.** Risk perception mapping among disaster relief nurses.

Additionally, perceptions across the age groups were analyzed for the 15 risk factors for "unknown risk". Notably, the perception of the "X-ray test" was significantly different between the age groups ($p = 0.009$; Table 3).

**Table 3.** Differences in "unknown risk" scores by age group (mean ± standard deviation).

| Age Group | 20s (n = 9) | 30s (n = 105) | 40s (n = 179) | 50s (n = 91) | ≥60s (n = 17) | *p*-Value |
|---|---|---|---|---|---|---|
| Radiation Exposure | 4.22 ± 1.23 | 4.35 ± 1.68 | 3.94 ± 1.91 | 4.12 ± 1.60 | 4.29 ± 1.27 | 0.289 |
| Volcanic Eruption | 3.44 ± 1.64 | 4.10 ± 1.64 | 3.77 ± 1.69 | 3.92 ± 1.32 | 4.24 ± 1.31 | 0.472 |
| Flood | 3.33 ± 1.64 | 4.32 ± 1.64 | 3.88 ± 1.70 | 4.13 ± 1.49 | 4.71 ± 1.07 | 0.69 |
| Norovirus | 3.22 ± 1.57 | 3.35 ± 1.57 | 3.00 ± 1.75 | 3.34 ± 1.63 | 2.65 ± 1.61 | 0.618 |
| Radioactive Contamination | 4.56 ± 1.67 | 4.27 ± 1.67 | 4.01 ± 1.88 | 4.25 ± 1.52 | 4.06 ± 1.39 | 0.501 |
| Nuclear Weapons (Nuclear War) | 4.56 ± 1.78 | 4.28 ± 1.78 | 4.22 ± 2.09 | 4.53 ± 1.69 | 4.65 ± 1.53 | 0.345 |
| Mass Infection | 4.22 ± 1.59 | 4.24 ± 1.59 | 3.97 ± 1.73 | 4.15 ± 1.55 | 4.47 ± 1.42 | 0.836 |
| Ebola | 4.67 ± 1.65 | 4.11 ± 1.65 | 3.88 ± 1.94 | 4.08 ± 1.54 | 4.18 ± 1.54 | 0.636 |
| Car Accident | 3.89 ± 1.64 | 3.50 ± 1.64 | 3.20 ± 1.73 | 3.35 ± 1.37 | 3.24 ± 1.26 | 0.319 |
| X-ray Test | 3.56 ± 1.81 | 3.17 ± 1.81 | 2.28 ± 1.73 | 2.67 ± 1.51 | 2.12 ± 1.02 | 0.009 |
| Dengue Fever | 4.22 ± 1.64 | 3.85 ± 1.64 | 3.49 ± 1.86 | 3.62 ± 1.47 | 3.41 ± 1.72 | 0.133 |
| Cervical Cancer Vaccine | 4.33 ± 1.57 | 3.20 ± 1.57 | 2.80 ± 1.71 | 3.32 ± 1.27 | 2.76 ± 1.48 | 0.539 |
| Nuclear Reactor Accident | 5.00 ± 1.79 | 4.35 ± 1.79 | 4.11 ± 1.94 | 4.52 ± 1.67 | 4.88 ± 1.18 | 0.578 |
| Tobacco | 3.67 ± 1.67 | 3.07 ± 1.67 | 2.93 ± 1.73 | 3.40 ± 1.53 | 3.24 ± 1.44 | 0.216 |
| Genetic Engineering | 4.22 ± 1.66 | 3.98 ± 1.66 | 3.55 ± 1.80 | 3.96 ± 1.44 | 3.59 ± 1.24 | 0.439 |

## 4. Discussion

In this study, we surveyed DRNs who were registered in six Japanese prefectures to gain insights into their perceptions of 15 risk factors. Utilizing this data, we developed a risk perception map to illustrate how perceptions vary across age groups. This approach allowed the examination of DRNs' nuanced understanding of risks, which is crucial for both the training of DRNs and their effectiveness in disaster response scenarios. Notably, the psychological impact of risk perception among healthcare professionals can be significant and affects not only the mental well-being but also the performance of these professionals in high-stress scenarios [17]. Therefore, understanding the nuances of risk perception among DRNs is not only an academic exercise but a practical necessity. Our risk perception map reveals a pattern consistent with that in Slovic's seminal 1987 report on risk perception [5]. This consistency over several decades underscores the enduring nature of how risks are psychologically processed among not only the general population but also healthcare professionals such as DRNs. This consistent pattern also lends further credence to our study and highlights the importance of continually updating risk perception models to inform training and policies in healthcare settings. Our study further raises important questions about the role of institutional policies and training programs in shaping risk perceptions among DRNs. For instance, the findings challenge whether current training modules adequately address the psychological aspects of risk, especially in high-stress environments such as disaster zones. This issue is particularly relevant given that risk perception has been shown to influence psychological distress among healthcare workers [17]. Future studies should also explore the impact of organizational culture—a potential key influencing factor in disaster response effectiveness—on DRNs' risk perceptions.

### 4.1. Radiation-Related Risks

In this study, the DRNs perceived radiation- and nuclear-related risks as high. The finding of a heightened perception of radiation and nuclear-related risks is consistent with that of previous surveys [5,18], suggesting that DRNs may face challenges when managing nuclear disasters. This heightened perception of risk could affect the confidence of DRNs in barrier measures such as the use of protective gear, especially when addressing radiation and nuclear-related risks [19].

Conversely, consistent with previous studies, DRNs perceived radiography as low-risk [13]. This heightened perception suggests that managing nuclear disasters may be challenging for DRNs. Consequently, specialized training modules that focus on radiation safety and psychological resilience are imperative. These modules could demystify the complex risks associated with nuclear disasters and equip DRNs with the tools to effectively

manage high-stress situations. Insights from past nuclear disasters such as the Fukushima accident further emphasize the need for targeted interventions in risk communication strategies that nurses should implement [20]. After the Fukushima accident, the nursing profession reported feelings of extraordinary powerlessness [21], presumably because of the high perceived radiation risk. Reducing nurses' anxiety about radiation risk will likely lead to satisfactory disaster relief practices. Given these concerns, it is imperative to incorporate specialized training modules that focus on radiation safety and psychological resilience, as suggested by recent studies [22]. These modules should include components of Incident Management Systems (IMS) and disaster triage to better prepare DRNs for nuclear disasters. Given the high perceived susceptibility to risks such as COVID-19 infection among healthcare professionals [23], balancing the provision of quality care to patients with the responsibility and ability to protect DRNs is crucial. This approach may involve strategies that empower and improve DRNs' knowledge and psychological support mechanisms. Moreover, the ethical implications of our findings cannot be overlooked. If younger nurses perceive certain procedures as high-risk because of a lack of experience or education, there is an ethical obligation to ensure that training programs are robust enough to equip these nurses with the necessary skills and confidence to perform their duties effectively [24].

### 4.2. Mass Infection Risk Amid the COVID-19 Pandemic

We further incorporated risk perception items related to "mass infection" in light of the ongoing COVID-19 pandemic. Both "dread risks" and "unknown risks" were perceived as high, indicating that DRNs, who typically work in hospitals but are dispatched to disaster areas when needed, were heavily engaged with COVID-19. Given the high levels of social and scientific uncertainty surrounding the virus at the time of the survey, the DRNs likely also feared personal infection and thus perceived mass infection as high-risk [25–27]. This acute awareness calls for a shift in the focus of disaster nursing, which should include not only curriculum updates but also enhanced psychological support mechanisms for DRNs. Notably, the COVID-19 pandemic has accelerated the need for digital health literacy among DRNs. The pandemic has shown that understanding and interpreting online health-related information is crucial for informed decision making among healthcare professionals including DRNs [28]. Given the heightened risk perception surrounding mass infections, it is crucial to enhance psychological support mechanisms for DRNs. Strategies could include stress management and coping skills training, underpinned by the positive correlation between training needs and job satisfaction [29].

By delving into these two critical areas, our study provides a comprehensive framework for risk assessment. Our findings guide targeted training and policy interventions and enhance our collective disaster preparedness and response capabilities.

### 4.3. Risk Perception Differences by Age

The age-based analysis of the participants revealed that the younger individuals felt more dread about the risk of mass infection than their older peers. Similarly, younger respondents felt more uncertainty about the "X-ray test". As of 8 August 2023, COVID-19 has infected 768,983,095 people globally and caused 6,953,743 deaths [30]. In Japan, the infection and mortality counts stand at 33,803,572 and 74,694, respectively [31]. In this study, the DRNs, who typically work in hospitals and other medical facilities, may have been directly involved in the COVID-19 response. The reports of burnout among nurses who respond to COVID-19 [32] suggest that those who treat infected patients experience considerable stress. Furthermore, given the potential risk of personal infection, nurses may have categorized this risk as a "dread risk". Finally, the $p$-value of 0.056 for "nuclear reactor accident" was marginally above the significance level. Although not statistically significant, this result could be indicative of a trend that warrants further investigation. Given the limitations of our sample size, we cannot rule out the possibility that a β error affected this result. The marginally high $p$-value for "nuclear reactor accident" suggests

that this area requires further study to allow a full understanding of the implications for DRN training and preparedness—especially in the context of previous nuclear disasters such as the Fukushima accident.

Younger nurses may have perceived the "X-ray test" as a high-risk procedure because of their limited clinical experience. Conversely, older nurses who are accustomed to radiography because of regular exposure during their routine duties may have perceived radiography as low-risk. The age-based differences in risk perception regarding the "X-ray test" may have been influenced by a variety of factors beyond mere clinical experience. For example, younger nurses may be more exposed to recent educational curricula that emphasize the risks associated with radiography, thereby affecting their risk perception. This finding is in line with the notion that differing work areas and roles within healthcare influence risk perception [33]. Conversely, older nurses who have been in the field longer than their younger peers may have developed coping mechanisms or heuristic approaches to manage the stress and perceived risks associated with radiography. Older nurses' long-term exposure to the procedure, coupled with a lack of adverse outcomes, may contribute to a form of "risk habituation", in which the perceived risk decreases over time because of familiarity [33]. Moreover, the generational gap may play a role: older nurses may belong to a generation in which the risks associated with radiography were not as heavily emphasized, leading to a more relaxed perception of the procedure. In contrast, younger nurses, who are part of a more risk-averse generation, may perceive the same procedure as high-risk because of societal shifts in the understanding and communication of medical risks [34]. Finally, our study opens avenues for future research. A potential area of focus could be the development of targeted interventions that modify risk perceptions among DRNs, thereby improving their psychological resilience and overall effectiveness in disaster management [35].

### *4.4. Strengths and Limitations*

Although insightful, our study has certain limitations. The sampling of DRNs from only six prefectures in Japan may have restricted the generalizability of our findings to the broader national DRN population. The cross-sectional design of the research allowed the capture of perceptions at a singular time point, limiting the ability to understand evolving perceptions or infer causative relationships. Moreover, potential influencing factors may not have been eliminated. Although we acknowledge that the reliance on self-reported data introduces biases such as recall and social desirability, we were unable to conduct sensitivity analyses or pre-tests to quantify its impact. This is a study limitation and should be considered when interpreting the results. Additionally, the 47.2% response rate indicates the possibility of a non-response bias; we were unable to assess the impact of this potential bias on our findings. Future research should employ methods that can quantify or mitigate these biases. Despite our extensive focus on 15 risk factors, we may not have considered other significant risks or influential variables. Future research should consider the efficacy of Competency-Based Training (CBT) in improving various aspects of novice nurses' performance, as indicated by existing studies [36]. Such an approach could offer a more comprehensive training protocol that addresses both technical and psychological aspects of disaster response. Conversely, our study has several strengths. The focus on DRNs in Japan offers a deep exploration of this vital group that is essential in disaster management. Conducted during the challenging COVID-19 pandemic period, our study provides timely insights into risk perceptions that were gathered during an unparalleled global health crisis. The comprehensive assessment of 15 risk factors provides a well-rounded perspective on risk perceptions among DRNs. Finally, the study design, which builds upon foundational research, ensures that our contributions are both innovative and rooted in the established knowledge in the field.

## 5. Conclusions

Our study provides far-reaching insight into the risk perceptions among DRNs in Japan. The results, which are consistent with those of prior studies, emphasize the long-standing nature of certain risk perceptions, such as those of radiation. However, newer global challenges such as the COVID-19 pandemic have given rise to fresh perspectives and have highlighted the dynamic nature of risk perception, which changes with evolving global contexts. Younger DRNs who may have limited exposure to certain procedures such as radiography have a higher risk perception of these procedures, underscoring the importance of experience in shaping perceptions. Addressing these perceptions to ensure that DRNs are adequately prepared and supported is imperative and can ultimately enhance disaster response effectiveness in Japan.

**Author Contributions:** Conceptualization, A.N. and T.Y.; methodology, A.N. and T.Y; software, T.Y.; validation, Y.Y.; formal analysis, T.Y.; investigation, A.N., H.U. and T.S.; resources, H.U.; data curation, T.Y.; writing—original draft preparation, A.N. and T.Y.; writing—review and editing, Y.Y., H.U., T.S. and Y.M.; visualization, T.Y.; supervision, T.Y. and T.S.; project administration, H.U.; funding acquisition, Y.M. All authors have read and agreed to the published version of the manuscript.

**Funding:** This study obtained funding from the Research and Education Center for Natural Hazards, Kagoshima University.

**Institutional Review Board Statement:** After receiving all the necessary information about the study and providing informed consent written in the research cooperative documents, the participants were asked to complete the questionnaire. This study was conducted in accordance with the Declaration of Helsinki and approved by the Ethics Committee of Nagasaki University Graduate School of Biomedical Sciences (approval number: 20032702).

**Informed Consent Statement:** Informed consent was obtained from all participants involved in the study.

**Data Availability Statement:** All data are available from the corresponding author on reasonable request.

**Public Involvement Statement:** No public involvement in any aspect of this research.

**Guidelines and Standard Statement:** This manuscript was drafted against the Strengthening the Reporting of Observational Studies in Epidemiology guidelines for cross-sectional research.

**Acknowledgments:** We thank all study participants and Anahid Pinchis for editing a draft of this manuscript.

**Conflicts of Interest:** The authors declare no conflict of interest.

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
