# Peer review of "How Do Disaster Relief Nurses in Japan Perceive and Respond to Risks? A Cross-Sectional Study"

_nursrep, doi:10.3390/nursrep13040118_

Round 1

Reviewer 1 Report

This study examined the risk perceptions among disaster relief nurses (DRNs) in Japan by focusing on risk factors associated with frequent natural disasters and the COVID-19 pandemic. The comments are as follows.

1. Besides Japan, are there similar studies focus on risk perceptions among healthcare workers in other countries/regions? If yes, the introduction part should include these studies.

2. In the study design, have the authors considered and calculated sample size before distributing questionnaire? Is “986 nurses” has covered all DRNs in the selected areas and healthcare agencies? The authors need to note and explain in the manuscript. Besides, the criteria for inclusion and exclusion should also be noted.

3. In the statistical methods, Shapiro–Wilk test and Jonckheere–Terpstra test should be briefly explained with reference.

4. Figure 2 can be improved as some words have been shaded with lines.

5. Table 1 and relevant description should be relocated to results part.

6. In Table 2, P-values for X-ray test and Nuclear Reactor Accident exhibited a marginal statistical association (between 0.05 and 0.1). I suggest the authors can interpret it in the results and also explain them in the discussion.

7. The written language need to be improved, and errors need to be corrected.

N/A

Reviewer 2 Report

While the study offers critical insights, there are specific areas I believe could benefit from further attention to enhance the depth and clarity of the presented work:

The research focuses on DRNs from only six prefectures in Japan. While these regions might have specific significance or relevance, this selection potentially narrows the study's overall applicability. Expanding the scope or offering a detailed rationale for the chosen prefectures might improve the study's external validity and address concerns about its broader generalizability.

The cross-sectional nature of your study presents a snapshot of perceptions at one particular moment. Given the rapidly evolving nature of disasters and their subsequent impacts, this design may not capture dynamic changes in perceptions over time. It might be beneficial to highlight this limitation more prominently and discuss the potential implications it has for the interpretation of your results.

The reliance on self-reported data, as you've acknowledged, opens the door to various biases, including recall and social desirability biases. The reported response rate of 47.2% further suggests the potential for non-response bias. Delineating the potential implications of these biases on your findings and possibly providing strategies employed to mitigate them would strengthen your methodology.

While you've acknowledged some limitations, a more in-depth and structured limitations section could be invaluable. By comprehensively addressing potential weaknesses and their implications, readers will be better positioned to contextualize the study's findings within its bounds.

The discussion section offers a vital space to extrapolate on the broader implications and relevance of your findings. The current section appears somewhat concise. Elaborating more on how your findings fit within broader narratives, trends in the field, or potential applications could enhance the overall contribution of your work.

A richer array of references could bolster your article's claims and provide readers with a broader landscape of the existing literature. Diversifying your references to include a wider range of perspectives and research methods might also enhance the depth and breadth of your study's context.

I trust these suggestions are received in the constructive spirit they are intended. Your work contributes significantly to our understanding of DRNs' risk perceptions, especially within the Japanese context. I believe that with these enhancements, the article's impact and reach will only further solidify.

Warm regards,

Round 2

Reviewer 1 Report

The authors have extensively revised the manuscript according to the reviewer's comments. Thus, the manuscript can be accepted for publication after proofreading.

It needs minor English language revison and proofreading.

Author Response

Dear Reviewer 1

Thank you for your valuable comment on this review.

Sincerely

Reviewer 2 Report

Although the authors have refined the paper, it still necessitates the inclusion of specific recommendations, such as training protocols or solutions for nurses. I recommend to add some based on references:

https://doi.org/10.1371/journal.pone.0244488

https://doi.org/10.1371/journal.pone.0277484

https://doi.org/10.2147/RMHP.S312940

https://doi.org/10.1016/j.nepr.2022.103327

Author Response

Dear Reviewer 2

Thank you for your valuable comment on this review.

We added the three articles you suggested into our manuscripts (line 221-225, 247-250, 310-314).

Sincerely